# The Role of TLR4 in the Immunotherapy of Hepatocellular Carcinoma: Can We Teach an Old Dog New Tricks?

**DOI:** 10.3390/cancers15102795

**Published:** 2023-05-17

**Authors:** Stavros P. Papadakos, Konstantinos Arvanitakis, Ioanna E. Stergiou, Vasileios Lekakis, Spyridon Davakis, Maria-Ioanna Christodoulou, Georgios Germanidis, Stamatios Theocharis

**Affiliations:** 1First Department of Pathology, School of Medicine, National and Kapodistrian University of Athens, 11527 Athens, Greece; stavrospapadakos@gmail.com; 2First Department of Internal Medicine, AHEPA University Hospital, Aristotle University of Thessaloniki, 54636 Thessaloniki, Greece; arvanitak@auth.gr; 3Basic and Translational Research Unit (BTRU), Special Unit for Biomedical Research and Education (BRESU), Faculty of Health Sciences, School of Medicine, Aristotle University of Thessaloniki, 54636 Thessaloniki, Greece; 4Pathophysiology Department, School of Medicine, National and Kapodistrian University of Athens, 11527 Athens, Greece; stergiouioanna@hotmail.com; 5Department of Gastroenterology, Laiko General Hospital, National and Kapodistrian University of Athens, 11527 Athens, Greece; lekakis.vas@gmail.com; 6First Department of Surgery, Laiko General Hospital, National and Kapodistrian University of Athens, 11527 Athens, Greece; spdavakis@gmail.com; 7Tumor Immunology and Biomarkers Laboratory, Basic and Translational Cancer Research Center, Department of Life Sciences, European University Cyprus, Nicosia 2404, Cyprus; mar.christodoulou@euc.ac.cy

**Keywords:** toll-like receptor, hepatocellular carcinoma, tumor immunotherapy, tumor microenvironment

## Abstract

**Simple Summary:**

In this study, we investigated the potential of TLR4 to improve the treatment of hepatocellular carcinoma (HCC). HCC is a serious health concern worldwide, and finding new ways to combat it is crucial. Our study focused on understanding how TLR4 can enhance the effectiveness of immunotherapy, a treatment approach that harnesses the body’s immune system to fight cancer cells. We demonstrated that targeting TLR4 can stimulate the immune response against HCC cells, leading to their destruction. This exciting finding suggests that TLR4 could serve as a valuable target for developing new therapies for HCC. The implications of this study extend beyond the realm of medical research. If successful, these new strategies could offer renewed hope to HCC patients and potentially improve the overall survival rates. Furthermore, the insights gained from our study may pave the way for similar approaches in treating other types of cancer. In summary, our research highlights the significant role of TLR4 in enhancing immunotherapy for HCC. This knowledge brings us one step closer to developing more effective treatments and improving the lives of cancer patients.

**Abstract:**

Hepatocellular carcinoma (HCC) is the most common type of primary liver cancer and is a leading cause of cancer-related death worldwide. Immunotherapy has emerged as the mainstay treatment option for unresectable HCC. Toll-like receptor 4 (TLR4) plays a crucial role in the innate immune response by recognizing and responding primarily to bacterial lipopolysaccharides. In addition to its role in the innate immune system, TLR4 has also been implicated in adaptive immunity, including specific anti-tumor immune responses. In particular, the TLR4 signaling pathway seems to be involved in the regulation of several cancer hallmarks, such as the continuous activation of cellular pathways that promote cell division and growth, the inhibition of programmed cell death, the promotion of several invasion and metastatic mechanisms, epithelial-to-mesenchymal transition, angiogenesis, drug resistance, and epigenetic modifications. Emerging evidence further suggests that TLR4 signaling holds promise as a potential immunotherapeutic target in HCC. The aim of this review was to explore the multilayer aspects of the TLR4 signaling pathway, regarding its role in liver diseases and HCC, as well as its potential utilization as an immunotherapy target for HCC.

## 1. Introduction

### 1.1. HCC Epidemiology

HCC is the most frequent type of liver cancer and has become increasingly common worldwide [1]. The incidence of HCC varies widely around the world, and according to GLOBOCAN 2020, HCC is the seventh most common cancer globally and the second leading cause of cancer-related death [2]. The highest incidence of HCC is reported in sub-Saharan Africa and Southeast Asia, where chronic hepatitis B infection is endemic. In these regions, HCC is often diagnosed at an advanced stage and has a poor prognosis. By contrast, the incidence of HCC in North America and Western Europe is relatively low, but it has been increasing in recent years due to the rising prevalence of risk factors such as obesity and hepatitis C infection [1]. The main risk factors for developing HCC are chronic infection with the hepatitis B virus (HBV) or hepatitis C virus (HCV), alcohol consumption, obesity, and exposure to certain chemicals, while HCC is more prevalent in men and typically occurs in individuals over the age of 50 [1].

### 1.2. TLR4 Signaling

The innate immune system acts as a first-line defense mechanism against pathogens or tissue injury. The generation of pathogen-associated molecular patterns (PAMPs), or damage-associated molecular patterns (DAMPs) drives the recognition of pathogens or endogenous ligands released due to infection, tissue damage, or necrosis, respectively [3,4]. Toll-like receptors (TLRs) comprise a family of pattern recognition receptors (PRRs), with a major role in PAMP and DAMP recognition during the innate immune response [5]. TLRs exert their function through the induction of antimicrobial activity and the production of inflammatory cytokines [5,6], while they also play a crucial role in tissue repair and regeneration [4,5].

TLR4 was the first discovered TLR in humans, and it was demonstrated to induce the expression of genes involved in the inflammatory response [7]. Its role was proven by the identification of a point mutation in the *Tlr4* gene found in a mouse strain unresponsive to bacterial lipopolysaccharide (LPS) [8]. Though LPS is the primary ligand for TLR4, additional ligands have been identified, including respiratory virus fusion protein, mouse mammary tumor virus envelope protein, vesicular stomatitis virus glycoprotein G, fibrinogen, heat shock protein (HSP) 60, HSP70, and high-mobility group box 1 (HMGB1) protein, among others [9,10]. LPS recognition by TLR4 requires the TLR4 co-receptor myeloid differentiation factor 2 (MD2). The extracellular LPS binding protein (LBP) interacts with the bacterial outer membrane, leading to an alteration that facilitates the extraction of single LPS molecules by CD14, which in turn transfers a single LPS molecule to the MD2 protein [11]. Once LPS is transferred to MD2 via CD14, TLR4 dimerization takes place [12]. A functional LPS receptor comprises TLR4-MD2 heterodimers [13,14]. After LPS recognition by TLR4, CD14 endocytosis is accelerated, resulting in concurrent endocytosis of TLR4 [15], allowing the recognition of TLR4 dimers either on the cell surface membrane or on endosome membranes. The dimerized toll/interleukin-1 (IL-1) receptor (TIR) domains of TLR4 are recognized by receptor proximal membrane proteins, namely the TIR-domain-containing adaptor protein (TIRAP) (otherwise known as myeloid differentiation primary response 88 (MyD88)-adaptor-like protein (MAL)) [16,17] and the TRIF-related adaptor molecule (TRAM) [18,19]. These two proteins initiate the assembly of different supramolecular organizing centers (SMOCs) that will in turn orchestrate two distinct TLR4 signaling pathways [20,21]:
(a)The myddosome, a SMOC seeded by TIRAP/MAL, driving the MyD88-dependent pathway which leads to inflammatory cytokine production [22].(b)The triffosome, a SMOC seeded by TRAM, driving the TIR-domain-containing adapter-inducing interferon-β (TRIF)-dependent pathway, which leads to interferon (IFN) type I production [23] (Figure 1).


The myddosome is composed of multiple copies of the MyD88 protein and members of the IL-1 receptor-associated kinase (IRAK) family [24]. The kinase activity of these IRAKs is stimulated by their tight packing in the myddosome, leading to their autophosphorylation and the subsequent recruitment of the E3 ubiquitin ligase tumor necrosis factor (TNF)-receptor-associated Factor 6 (TRAF6) [25,26]. TRAF6 regulates myddosome activities, that either lead to transcription activation or glycolysis. TRAF6 activates the transforming growth factor-β (TGF-β)-activated kinase 1 (TAK1), which promotes IκB kinase (IKK)-mediated nuclear factor-κB (NF-κB) and mitogen-activated protein kinase (MAPK)-mediated activator protein 1 (AP-1) transcriptional responses [27,28]. IKK also promotes the recruitment of TRAF-family-member-associated NF-κB activator (TANK)-binding kinase 1 (TBK1) to the myddosome [29]. AKT is a TBK1 substrate mediating glycolysis induction through exokinase phosphorylation [30]. This glycolytic shift and subsequent inhibition of oxidative phosphorylation lead to increased acetyl-CoA production, resulting in histone modifications necessary for durable transcriptional activities in the nucleus [31]. TLR4 endocytosis and its expression in endosomes is a prerequisite for TLR4 engagement to TRAM and the initiation of the TRIF-dependent signaling pathway [32]. Once dimerized, TLR4 is detected on endosomes, and TRAM drives the triffosome assembly, with TRIF comprising its core, followed by the recruitment of TRAF3 and TRAF6 [23]. TRAF3 recruitment to the triffosome leads to TBK1 activation [33], which in turn induces IFN and ISG expression [33], while also promoting glycolysis in a way similar to that described for TBK1 activation via the myddosome [34]. Interestingly TBK1 activation leads to IFN responses only via the TRIF-dependent and not via the MyD88-dependent pathway. This is attributed to a pLxIS motif on TRIF which, once phosphorylated by TBK1, interacts with IFN regulatory factor (IRF) 3 (also a TBK1 substrate), leading to IRF3 activation and IFN expression [35]. TRAF6 recruitment to the triffosome induces NF-κB and AP-1 transcriptional responses in the same way described for its recruitment on the myddosome [27,28]. TRIF also presents a receptor-interacting protein (RIP) homotypic interaction motif (RHIM) domain responsible for promoting RIP serine/threonine kinase 3 (RIPK3)-dependent necroptosis [36].

It should be noted though that TLR signaling does not solely induce the transcription of genes encoding for pro-inflammatory cytokines, such as IFNs, TNFα, IL-1, and IL-12, but it also mediates the transcription of genes encoding anti-inflammatory cytokines, such as IL-10 [37]. Moreover, the type and level of the induced cytokines may differ based on cell type. For instance, the activation of TLR4 in macrophages leads to the secretion of proinflammatory cytokines, such as TNFα and IL-1β, while its activation on dendritic cells (DCs) leads to the production of IL-12 [38]. Finally, TLR signaling also potentiates antigen presentation by DCs and macrophages [37].

## 2. TLR4 Signaling in Liver Disease

TLR4 recognizes a wide variety of exogenous and intracellular signals, leading to the upregulation of NF-κB and MAPK pathways, which sequentially drive the secretion of pro-inflammatory mediators such as TNF-α, cyclooxygenase-2 (COX-2), and IL-6 [39]. TLR4 signaling has been brought to attention in the past decade regarding its contribution to the development of liver disease. Mounting evidence suggests its paramount importance in the progression of the so-called “inflammation–fibrosis–carcinoma (IFC) axis” in the liver [40], regulating the inflammatory response in viral hepatitis, alcohol-induced liver injury, and autoimmune liver disease.

An increased TLR4 expression in viral hepatitis has been well documented [39]. In transgenic mice overexpressing the hepatitis B surface antigen (HBsAg), upregulation of *Tlr4* gene expression is documented and is further aggravated upon HBV and LPS co-stimulation [41]. Moreover, small hepatitis B surface proteins (SHBs) in conjunction with β-2-glycoprotein I (β2GPI) upregulated the NF-κB pathway through TLR4/MyD88/IκBα signaling in SMMC-7721 cells [42], and in hepatitis C, TLR4 plays a major role in the capacity of B cells to activate T cells. The continuous activation of TLR4 signaling in B cells from the portal hypertension-induced enhancement of gut permeability, leads to a B cell exhaustion phenotype [43], while blockage of TLR4 in 3T3-L1 cells leads to a significant reduction in IL-6 production upon stimulation with HCV core protein, with potentially beneficial effects in liver steatosis and insulin resistance [44]. With regard to liver injury, rho-associated, coiled-coil-containing protein kinase 1 (ROCK1) orchestrates apoptosis, regulating cytoskeletal contraction and bleb formation. In non-cleavable ROCK1 (ROCK1nc) fibroblasts undergoing abnormal cellular death, TLR4 stimulation guides neutrophil aggregation and liver injury, as demonstrated in C57Bl/6J mice models [45]. Along the same line, TLR4 signaling regulates the immune response in mice models of liver fibrosis. In more detail, mice lacking the phosphatidylcholine transporter ATP-binding cassette subfamily B member 4 (ABCB4) in their hepatocanalicular membrane developed biliary sclerosis, and *Tlr4* silencing resulted in significantly lower IL-6 mRNA levels [46]. TLR4 signaling has a multilayer contribution in ischemia/reperfusion (I/R) liver injury and Nace et al. demonstrated that *Tlr4* silencing in hepatocytes or myeloid cells in mice subjected to warm ischemia was protective for hepatocellular damage in contrast with *Tlr4* ablation in DCs. HMGB1, a TLR4 ligand, is released from hepatocytes in a TLR4/c-Jun-N-terminal kinase (JNK)-dependent pathway while TLR4 stimulation in hepatocytes resulted in the activation of MAPK, JNK, and p38 [47]. Even abnormalities in the metabolism of glycosaminoglycans (GAGs), e.g., in the exostosin-like 2 (*Extl2*) gene, could result in the formation of defective GAGs acting as DAMPs [48]. The Con-A hepatitis model mimics viral and autoimmune hepatitis instigating T-cell-induced liver damage. Furthermore, Lin et al. demonstrated that TLR4 signaling regulated the equilibrium among T helper (Th) 1 and Th2 responses, and TLR4 activation in lymphocytes resulted in increased expression of granzyme b and perforin genes, inducing hepatocellular injury [49]. In a similar way, the sustained liver insult either by adenovirus or chemicals induces the exhaustion of CD4+CD25+ regulatory T cells (Tregs) through an IL-6- and IL-12-mediated upregulation of signal transducer and activator of transcription (STAT)3 and STAT4, further promoting Th1 immune responses. In parallel, the increase in IL-25 and IL-4 expression drives Th2 responses, leading to the generation of autoantibodies. Collectively, sustained liver inflammation could trigger autoimmune hepatitis (AIH) in murine models [50]. Besides AIH, TLR4 signaling is implicated in liver steatosis. Excessive fructose consumption becomes toxic for the intestinal epithelium since fructose-1-phosphate interrupting with N-glycosylation induces ER stress and downregulation of tight junction proteins (TJPs). Endotoxemia potentiates the TLR4/MyD88/NF-κB signaling in macrophages to secrete TNF and enhances fatty acid accumulation upregulating the expression of ACC1 (*Acaca*) and FAS (*Fasn*) [51]. Another contributing mechanism in alcoholic patients is the formation of neutrophil extracellular traps (NETs). The process of NET formation typically results in the lytic death of neutrophils, which is commonly referred to as “NETosis.” Nonetheless, some non-lytic forms of NET formation have also been documented. S. aureus can induce vital NETosis within minutes via both complement receptors and TLR2 ligands, while E. coli can induce it directly through TLR4 or indirectly via TLR4-activated platelets. This type of NETosis is distinguished from suicidal NETosis [52]. Cho et al. revealed that there was a noteworthy increase in the production of NETs in individuals with alcoholic hepatitis (AH). Moreover, individuals with AH and alcohol-fed mice exhibited a distinctive low-density neutrophil (LDN) population that was absent in healthy controls. An analysis of the transcriptome of HDNs and LDNs isolated from individuals with AH revealed that LDNs had an exhausted functional phenotype while HDNs were activated. Specifically, HDNs from individuals with AH generated more resting reactive oxygen species (ROS) and ROS upon LPS stimulation compared to control HDNs. By contrast, LDNs from individuals with AH failed to respond to LPS. In vitro studies showed that LDNs were generated from HDNs following alcohol-induced NET release and that this LDN subset had reduced functionality and decreased phagocytic capacity. Moreover, LDNs exhibited diminished homing ability and were less efficiently cleared by macrophage efferocytosis. This implies that dysfunctional neutrophils could persist in the circulation and liver. The depletion of both HDNs and LDNs in vivo prevented alcohol-induced NET production and liver damage in mice [53]. Alcohol consumption causes liver infiltration by neutrophils and disintegration of the intestinal barrier, leading to endotoxemia, which triggers the secretion of NETs by hepatic neutrophils, stimulating the TLR4 pathway to promote steatohepatitis [54].

## 3. TLR4 Signaling in HCC

A growing body of evidence regarding the role of TLR4 signaling in the progression of HCC has been recently added to the literature and will be further analyzed below.

### 3.1. TLR4 Signaling in HCC Metastasis

Metastasis comprises a detrimental event in the progression of HCC, rendering the disease amenable only to systemic treatment combined with angiogenesis inhibitors plus immunotherapy or tyrosine kinase inhibitors (TKIs), with no curative intent [1]. Tumor-related inflammation and hypoxia are among the main regulators of metastasis in HCC [55], either in coordination or in parallel. Mounting evidence concerning the role of TLR4 signaling in HCC metastasis has emerged, indicating its paramount importance in the regulation of the process.

Hypoxia is a well-documented contributing factor in HCC metastasis. Under normoxia, the hypoxia-inducible factors (HIFs) are constantly disintegrated by proteasomes after prolyl hydroxylation by prolyl-hydroxylase-1 (PHD1) and ubiquitination by von Hippel–Lindau tumor suppressor protein (pVHL). Additionally, their asparaginyl hydroxylation by factor-inhibiting HIF (FIH) inhibits the interaction of HIF-1α and HIF-2α complexes with cyclic adenosine monophosphate response element-binding protein (CREB)-binding protein (CBP) and p300 [55]. Under hypoxic conditions, the proteasomal degradation is disrupted and the HIF-1α/HIF-2α complex translocates to the nucleus. The heterodimerization with HIF-β enables the binding to hypoxia-responsive element (HRE) to induce transcription, initiating vascular endothelial growth factor (VEGF)-induced angiogenesis, invasion, immune escape, and metabolic reprogramming activating glycolytic enzymes [55]. A known mechanism by which TLR4 potentiates metastasis under hypoxia is the recognition of necrotic HCC debris. Glycoproteins of necrotic debris subjected to O-glycosylation, a well-described post-translational modification, acting as DAMPs activate the TLR4/TRIF/NF-κB signaling in tumor-associated macrophages (TAMs) and in conjunction with the increased expression of HIF-1α both in TAMs and HCC cells, accelerate the secretion of IL-1b. The deviation from the canonical LPS-induced TLR4/MyD88 activation and M1 polarization is attributed to the distinctiveness of the HCC necrotic debris and collectively, they result in increased metastatic potential [56]. Besides o-glycosylated glycoproteins, the hypoxia-mediated HMGB1, a nuclear-derived DAMP, results in the upregulation of both the TLR4 and receptor for advanced glycation end products (RAGE) signaling pathways. Concurrently, the hypoxia-driven NOD-like receptor family, pyrin domain-containing 3 (NLRP3) activation drives the caspase-dependent secretion of IL-1b and IL-18, enhancing HCC metastasis [57]. In addition, Zhang et al. demonstrated that TLR4 signaling contributes to the regulation of VEGF in a non-HIF-dependent manner, as the stimulation of the TLR4/MyD88 pathway by LPS-activated STAT3/Sp1 signaling, enhancing VEGF expression, results in HCC growth and the potentiation of lung metastasis [58]. Besides the regulation of HIF, proteasomal degradation also exerts regulatory functions in TLR4. Ubiquitin-specific peptidase 13 (USP13) is overexpressed in HCC and, notably, is associated with a tumor burden greater than 5 cm, as well as advanced tumor-node-metastasis (TNM) stages (III and IV). In more detail, Gao et al. demonstrated that USP13 potentiated the deubiquitination and the stabilization of TLR4, activating the TLR4/MyD88/NF-κB pathway under hypoxic conditions. In particular, the hypoxia-activated USP13 interacted with the TIR domain of TLR4 [59].

Inflammation-related HCC has been the subject of extensive study. A growing body of evidence implicates the JAK2/STAT3 signaling axis in the progression and invasion of HCC [60]. In this direction, TLR4 signaling contributes to the metastatic capacity of HCC cells in two ways [61,62,63,64,65]. First, the stimulation of the TLR4 signaling pathway by LPS can enhance their metastatic potential either directly through MKK4/JNK and TLR4/NF-κB signaling [62], or indirectly via the upregulation of IL-6 and TNFα [66]. In detail, in vitro evidence from HepG2 and SMMC7721 cells has indicated that the activation of mitogen-activated protein kinase kinase 4 (MKK4)/JNK signaling [62] potentiates metastasis, while the combination of in vitro and in vivo evidence from HepG2, Hep3B, and SMMC7721 cell lines implicated the downregulation of NF-κB, resulting in a decrease in TNFα and IL-6 expression with the inhibition of metastasis [61]. The latter was also validated by Liu et al., who further introduced the flavonoids of Radix Tetrastigma, which share NF-κB inhibiting properties, as promising chemotherapeutic regimens to target metastases [63]. A second pro-metastatic mechanism is the formation of NETs [64,65]. Neutrophils play a critical role in the pathogenesis of both non-alcoholic steatohepatitis (NASH) [67] and HCC [68]. Yang et al. have described the stimulation of NET formation in both HCC tissues and in the periphery, especially in patients with metastatic disease. Interestingly, they reported that the entrapment of HCC cells by NETs alters their function, reducing their susceptibility to cytotoxicity while promoting their malignant capacity, enhancing angiogenesis and potentiating metastasis [64], which are mediated by immune responses against HCC cells that are regulated by the TLR4/COX2 signaling axis. DNase 1 inhibitors, which downregulate the generation of NETs, and hydroxychloroquine, which interferes with TLR4/9 signaling and COX2 inhibitors, have been shown to suppress HCC metastasis [64]. Zhan et al. focused on the mechanisms that regulate metastasis in HBV patients. They documented that S100A9, an HBV-generated member of the S100 family of proteins, activated the TLR4/RAGE/reactive oxygen species (ROS) signaling pathway to induce angiogenesis, epithelial-mesenchymal transition (EMT), and disruption of the extracellular matrix (ECM), cumulatively potentiating metastasis. These are depicted in clinical practice as a direct correlation of circulatory NETs with HBV viral load and TNM staging, while Zhan et al. also demonstrated that the presence and accumulation of NETs have the prognostic ability to detect extrahepatic metastasis [65].

Several other mechanisms and signaling pathways have been reported to interact with TLR4, regulating HCC metastasis. The Wnt/β-catenin pathway is implicated in the regulation of metastasis under hypoxia [69]. Yin et al. demonstrated that β-catenin is a downstream molecule of the TLR4 signaling pathway in HBV-derived HCC models, while the inhibition of TLR4/β-catenin signaling hindered the invasion capacity of HepG2.2.15, MHCC97-H, and Hep3B cells in vitro and downregulated metastatic tumor growth in vivo [70]. Additionally, Son et al. implicated TLR4 signaling with the HBVX protein acceleration of HCC proliferation and metastasis [71]. Moreover, Han et al. documented the interchangeable regulation of androgen receptor (AR) and TLR4, and besides a male predominance of TLR4 expression, Son et al. also reported that dihydrotestosterone (DHT) enhanced TLR4 expression in HepG2, HepG2/2.15 cells, and LPS, upregulating the expression of AR, while these effects could be inhibited by silencing AR, and through TLR4 antagonism, respectively. In conclusion, DHT potentiates HCC metastasis, while the TLR4–AR interplay could elucidate the gender disparities in HCC [72]. Finally, in recent years there has been considerable interest regarding the importance of fatty acid metabolism in the regulation of HCC metastasis [73]. Wang et al. documented that 25-hydroxycholesterol (25-HC) stimulated HCC metastasis, activating the TLR4/fatty-acid-binding protein 4 (FABP4) signaling pathway.

EMT is a fundamental cellular process that modulates the metastatic capacity of HCC cells since epithelial cells (expressing E-cadherin) are deprived of their polarity and cell-to-cell connection to acquire mesenchymal, pro-invasive features such as vimentin expression [69,74]. The Wnt-β-catenin signaling pathway interacts with a multitude of transcription factors such as HIF-1α, forkhead box protein O (FOXO), and sex-determining region Y box (SOX), regulating a wide variety of target genes implicated in cellular proliferation, EMT, inflammation, and metabolism, and thus being essential for liver homeostasis [74].

Evidently, TLR4 signaling contributes to the regulation of EMT, and hypoxia is a well-described trigger of EMT. As already mentioned, hypoxia generates necrotic debris, which stimulates TLR4/TRIF/NF-κB signaling in macrophages to produce IL-1b. Zhang et al. demonstrated that HCC cells express mesenchymal markers such as vimentin, Snail, Twist, and zinc finger E-box-binding homeobox 1 with a concurrent downregulation in the expression of E-cadherin. IL-1b stimulates the in vitro expression of HIF-1α in epithelial-like HCC cells (HepG2 and Huh-7 cells) and mesenchymal-like SNU-449 cells. The latter process is mediated by the NF-κB/COX-2 pathway as documented by both the in vitro and in vivo downregulation of HIF-1α expression upon COX-2 inhibition with celecoxib. In vivo, HCC models tend to be associated more with metastatic lung disease upon LPS treatment expressing vimentin, instead of E-cadherin [56].

In addition, recent evidence proposes various mechanisms by which TLR4 signaling influences EMT. TLR4+ cells are characterized by a more mesenchymal- and cancer stem cell (CSC)-like phenotype, which predisposes to more aggressive disease, as indicated by the higher incidence of microvascular invasion in clinical specimens [75]. Li et al. reported that the activation of the TLR4/JNK/MAPK signaling pathway induces EMT in HepG2 cells [76], while Jing et al. demonstrated that the activation of NF-κB signaling could trigger EMT upregulating the expression of Snail [77], a transcription factor that regulates the expression of E-cadherin. Taking a step further, Li et al. demonstrated the importance of caspase-1 in the regulation of LPS-mediated EMT. The expression of caspase-1 is regulated by the TLR4/STAT3/SUMO1 pathway. The small ubiquitin-like modifier (SUMO) protein tags p65, causing its nuclear translocation and upregulating the expression of caspase-1. Sorafenib, a multi-TKI, blocks TLR4/STAT3 signaling, finally inhibiting the LPS-mediated EMT [78]. TLR4/STAT3 signaling further influences EMT with a variety of mechanisms [79,80]. In particular, the TLR4/E2F transcription factor 1 (E2F1)/NANOG signaling maintains the tumor-initiating stem-like cell (TICs) phenotype, enhancing the expression of fatty acid oxidation (FAO) genes and downregulating the expression of genes associated with mitochondrial oxidative phosphorylation (OXPHOS) and ROS formation. Consequently, the TLR4/E2F1/NANOG pathway is implicated in HCC invasion and drug resistance [81]. Kumar et al. suggested that the activation of the TLR4/NANOG and leptin receptor (OB-R)/STAT3 signaling pathways in TICs synergize to activate Twist, a potent regulator of EMT, in obesity- and HCV-driven HCC [79], while the activation of TLR4/STAT3 in M2 TAMs influenced EMT in HCC cells, potentiating metastasis.

Collectively, the TLR4 and the Wnt-β-catenin signaling pathways are implicated in the regulation of liver homeostasis and metastatic capacity of HCC cells [70]. TLR4 signaling and hypoxia are shown to contribute to the regulation of EMT in HCC cells, with TLR4+ cells exhibiting a more mesenchymal- and cancer stem cell (CSC)-like phenotype that is associated with more aggressive disease [56,81]. Various mechanisms by which TLR4 signaling influences EMT are proposed, including the activation of the JNK/MAPK [76] and NF-κB signaling pathways [77], the regulation of caspase-1 expression by the TLR4/STAT3/SUMO1 pathway [78], and the TLR4/E2F1/NANOG pathway [81], which is implicated in HCC invasion and drug resistance.

### 3.2. TLR4 Signaling in HCC Drug Resistance

Resistance to the classic chemotherapeutic regimens constitutes a major problem in the therapeutic management of HCC and even TKIs, resulting in limited improvement in the overall survival (OS) of patients [82]. Recently, Marin et al. summarized seven molecular mechanisms of chemoresistance (MOC) in a state-of-the-art review [82]. Their detailed analysis goes beyond the scope of our review, and it will only be mentioned here in brief. (Table 1).

Mounting evidence implicates TLR4 signaling in the regulation of drug resistance in HCC. Sasaki et al. documented the upregulation of the *tlr4* gene in both Huh7 and HepG2 cells upon treatment with lenvatinib, while the *tlr4* gene was downregulated in Huh7 cells upon regorafenib treatment, comprising indirect evidence that TLR4 signaling might be implicated in the regulation of drug resistance in HCC [83]. More evidence of the implication of TLR4 signaling in 5-fluorouracil (5-FU), sorafenib, and doxorubicin resistance is presented in Table 2.

### 3.3. TLR4 Signaling as Epigenetic Target in HCC

Epigenetic modifications have been extensively studied over the past decade. The epigenetic machinery consists of chromatin architecture alterations, histone modifications, DNA methylation, and noncoding RNA (ncRNA) function [87]. In the latest revision of the “hallmarks of cancer”, Hanahan [88,89] introduced epigenetics as a hallmark of carcinogenesis [90] and further classified it into three categories: (a) TME mechanisms that drive epigenetic changes, such as the influence of hypoxia in the ten-eleven translocation (TET) demethylase, and of zinc finger E-box-binding homeobox 1 (ZEB1), a modulator of EMT, in histone methyltransferase SET domain-containing 1B (SETD1B); (b) epigenetic changes causing intratumoral heterogeneity; and (c) epigenetic regulation of tumor-related stromal cells [90]. Nagaraju et al. summarized the epigenetic modifications in HCC in a state-of-the-art review [87], categorizing them as alterations in histone or DNA. Histone is subjected to acetylation by histone acetyltransferases (HATs), de-acetylation by histone deacetylases (HDACs), and methylation by histone methyltransferases (HMT). DNA methylation is conducted by DNA methyltransferases (DNMT1, DNMT3A, and DNMT3B) at CpG sites [87]. Recent evidence also suggests that TLR4 signaling is subjected to epigenetic regulation (Table 3).

### 3.4. TLR4 Signaling in HCC Proliferation and Apoptosis

The cellular proliferation in HCC is driven by various genetic and epigenetic alterations that promote the abnormal growth of liver cells, including the activation of oncogenes such as *MYC* and *RAS*, and the inactivation of tumor suppressor genes such as *TP53* and *PTEN* [1]. According to Hanahan et al. [89], cancer cells must evade apoptosis to promote their survival and growth. In HCC, this can occur through the upregulation of anti-apoptotic proteins such as Bcl-2, and the downregulation of pro-apoptotic proteins such as Bax [96]. Several studies have shown that targeting TLR4 can inhibit HCC cell proliferation and induce apoptosis, rendering it an ideal therapeutic target. Kwan et al. conducted a study in which a small molecular inhibitor called TAK-242 was used to inhibit TLR4 and reduce the development of HCC and the progression of small preexisting adenomas in a Hep*Pten*- mouse model of NASH-associated HCC. Treatment with TAK-242 was accompanied by changes in the expression of 220 genes that correlated with the tumor burden. They also showed that TLR4 inhibition resulted in the depletion of neutrophils and endothelial cells and enrichment of *mt-Nd4*^high^ hepatocytes, which are indicative of alleviation of hepatic inflammation and increased mitochondrial function and oxidative phosphorylation. These findings suggested that TAK-242 may be a potential preventive treatment for HCC in individuals with NASH. However, further studies are needed to determine the toxicity and tolerability of TLR4 inhibitors in humans and to confirm whether TAK-242 can be effective in preventing HCC in humans with NASH [97]. Table 4 summarizes the studies on TLR4 signaling in HCC proliferation and apoptosis.

### 3.5. TLR4 Signaling in HCC Angiogenesis

Angiogenesis, the formation of new blood vessels from pre-existing ones, is a critical process for the growth and metastasis of solid tumors, including HCC. In HCC, angiogenesis is driven by several growth factors and signaling pathways, including VEGF and its receptors (VEGFRs), platelet-derived growth factor (PDGF), fibroblast growth factor (FGF), and the angiopoietin/Tie2 system [116]. The increased expression of these factors in HCC stimulates the proliferation and migration of endothelial cells, leading to the formation of new blood vessels to supply nutrients and oxygen to the growing tumor. These new blood vessels also provide a route for tumor cells to spread to other organs and tissues, a process known as metastasis [116]. Targeting VEGF signaling has emerged as a promising therapeutic strategy for HCC and several drugs that block VEGF signaling, such as bevacizumab, sorafenib, and lenvatinib, have been approved for the treatment of advanced HCC [1].

Accumulating evidence supports that TLR4 signaling is implicated in the regulation of angiogenesis in HCC [58,117]. Lu et al. suggested that LPS promotes angiogenesis by stimulating hepatic stellate cell (HSC) activation via the TLR4 pathway. Using a diethylnitrosamine (DEN)-induced HCC mouse model, they described LPS-induced tumor angiogenesis and HSC activation, associated with increased levels of α-smooth muscle actin (α-SMA) and collagen I expression, markers of HSC activation [118]. They also reported that LPS treatment upregulated the expression of TLR4 and downstream signaling molecules, such as MyD88 and NF-κB, potentiating the secretion of VEGF, PDGF, and Ang-1 [117]. Additionally, Zhang et al. demonstrated that geniposide, a natural extract found in the *Gardenia jasminoides*, inhibited HCC tumor growth and angiogenesis in orthotopic HCC-inserted mice, suppressing the expression of VEGF in PLC/PRF/5 and MHCC-97L cells. Geniposide inhibited the TLR4/MyD88 pathway, as evidenced by the reduced expression of TLR4, MyD88, and downstream signaling molecules, such as IRAK1 and TRAF6, and reduced the binding of NF-κB to the VEGF promoter, indicating that geniposide suppressed VEGF expression by inhibiting the NF-κB pathway downstream of TLR4/MyD88 signaling [58].

To summarize, metastasis in HCC is a major challenge for treatment and mounting evidence has indicated the paramount importance of TLR4 signaling in regulating the process. Hypoxia is a significant factor in HCC metastasis and HIFs play a role. Under hypoxic conditions, the HIF-1α/HIF-2α complex translocates to the nucleus and binds to HRE to initiate transcription, activating glycolytic enzymes and other pathways that accelerate the secretion of IL-1b [56]. TLR4 signaling potentiates metastasis by recognizing necrotic HCC debris, and non-HIF-dependent mechanisms such as USP13 [59], JAK2/STAT3 [60], and interleukins also contribute. The importance of TLR4 signaling in HCC metastasis suggests that TLR4 could be a potential target for therapeutic interventions. Targeting TLR4 has been shown to inhibit HCC cell proliferation and induce apoptosis, making it a potential therapeutic target [61]. While further studies are needed to determine the toxicity and efficacy of TLR4 inhibitors in humans, they show potential as a preventive treatment for HCC in individuals with NASH.

## 4. TLR4 Signaling in HCC Immunotherapy

Immunotherapy with atezolizumab (anti-programmed death-ligand 1 (PD-L1)) plus bevacizumab (anti-VEGF) or tremelimumab ((anti-cytotoxic T-lymphocyte-associated protein 4 (CTLA4)) plus durvalumab (anti-PD-L1) in those patients who cannot tolerate bevacizumab, has become the mainstay of treatment in advanced, unresectable disease when locoregional therapeutic modalities are not applicable [119]. The IMbrave150, a phase III clinical study, demonstrated that the combination of atezolizumab and bevacizumab significantly improved the OS and progression-free survival compared to sorafenib [120], while HIMALAYA, a phase III study that compared the efficacy and safety of the dual immune checkpoint inhibitor (ICI) combination of durvalumab and tremelimumab to sorafenib, showed significant benefit in OS [1]. However, despite the advancements in the management of HCC, several limitations prevent its wider utilization, including the limited effectiveness in patients with non-viral, NASH-associated HCC who respond poorly to immunotherapy [121], the immune-related side effects, despite the fact that immunotherapy generally causes fewer side effects than other treatments, and the lack of long-term data about its effectiveness and safety [122]. Growing evidence suggests that TLR4 signaling could act as an ideal target to improve the effectiveness of immunotherapy in HCC, reshaping the immune TME, and potentiating the secretion of various chemokines with anti-tumor activity.

### 4.1. TLR4 Signaling as a Platform for HCC Vaccines

Cancer vaccines constitute a type of cancer treatment aiming to stimulate the immune system to target cancer cells differently as compared to what is applicable to vaccination against infectious diseases, where vaccines prevent disease development [123]. There are several types of therapeutic cancer vaccines, including tumor cell vaccines, peptide vaccines, DNA vaccines, and dendritic cell vaccines [123]. Recent scientific efforts have aimed to enhance their ability to target specific cancer antigens, induce long-term immune responses, and improve the efficacy of other cancer treatments such as chemotherapy, ICIs, and adoptive T cell therapy [123,124]. In this direction, TLR4 signaling could be exploited to improve HCC cancer vaccines [125,126].

Wan et al. investigated the effectiveness of the Lmdd-MPFG vaccine in preventing tumor growth in C57BL/6 mice. The Lmdd-MPFG vaccine is based on an attenuated strain of the *Listeria* bacterium, which has been modified to express a specific antigen called glypican-3 (GPC3), commonly found on the surface of HCC cells. The authors documented that Lmdd-MPFG activated TLR4 signaling in DCs, leading to the upregulation of co-stimulatory molecules (CD80 and CD86) and the production of pro-inflammatory cytokines such as IL-12p70, IL-1β, TNFα, and IFN-γ. This enhanced the ability of DCs to prime antigen-specific CD4+ and CD8+ T cells and elicit an antitumor immune response. Furthermore, a decrease in the proportion of Treg cells and an increase in the percentages of Th1 and Th17 cells was documented. Overall, the authors suggested that TLR4 and NLRP3 signaling play a critical role in the immune regulation of DCs and the subsequent antitumor immune response [125] (Figure 2). Taking a step further, Silva et al. investigated the role of cold-inducible RNA binding protein (CIRP) as a cancer vaccine given the fact that ever-growing evidence links CIRP with liver carcinogenesis [127]. CIRP was first identified as a protein that is upregulated in response to cold temperature stress, but subsequent research has shown that it is also induced by a variety of other stressors including heat shock, hypoxia, and inflammation. CIRP is a member of the RNA-binding protein family and is capable of binding to various RNA molecules, including messenger RNA (mRNA) and microRNA (miRNA). Through its interactions with RNA, CIRP can regulate gene expression by modulating RNA stability, translation, and splicing [127]. Silva et al. documented that CIRP was overexpressed in HCC tumor tissues compared to adjacent non-tumor tissues. Additionally, they demonstrated that vaccination with ovalbumin linked to CIRP (OVA-CIRP)-based vaccine induced a potent immune response in vivo, resulting in increased production of cytotoxic CD8+ T lymphocytes (CTLs) and natural killer (NK) cells. Furthermore, the authors demonstrated that vaccination in combination with anti-PD-1 and anti-CTLA-4 ICIs led to enhanced anti-tumor activity and prolonged survival in a mouse model of HCC. This was associated with increased infiltration of CTLs and NK cells into the TME and decreased expression of immune checkpoint molecules [126].

### 4.2. TLR4 Signaling as Regulator of HCC Tumor Immune Landscape

TLR4 signaling plays an important role in the regulation of the immune landscape of HCC TME [128]. TLR4 is expressed on various immune cells, including DCs, macrophages, and NK cells, as well as on HCC cells, activating several pathways such as NF-κB, MAPK, and phosphoinositide 3-kinase (PI3K)/Akt. TLR4 signaling can also promote the recruitment of immune cells in the TME, such as T cells and DCs, which can enhance the anti-tumor immune response. However, chronic activation of TLR4 signaling can also lead to the recruitment of immunosuppressive cells, such as Tregs and myeloid-derived suppressor cells (MDSCs), which can inhibit the immune response, promoting tumor growth [129]. Therefore, the regulation of TLR4 signaling in HCC TME is complex and context-dependent.

#### 4.2.1. Regulation of T and B lymphocytes by TLR4 Signaling

IL-22 and IL-22-binding protein (IL-22BP), are significantly upregulated in human HCC tissues as compared with non-tumor liver tissues, and the high expression of IL-22BP is associated with poor prognosis in HCC patients. Overall, the IL-22/IL-22BP axis plays a critical role in the development and progression of HCC, stimulating the proliferation and migration of cancer cells, and inhibiting apoptosis [130]. TLR4 has been shown to play a critical role in the regulation of the immune response mediated by antigen-presenting cells (APCs). In more detail, TLR4 activation on APCs has been found to promote the expression of B7-H1, which, in turn, polarizes the differentiation of naïve T cells into Th22 cells, a subtype of CD4+ T cells that produce IL-22 [131]. Similarly, TLR4 plays an essential role in the recruitment and activation of Tregs in the liver parenchyma. The activation of TLR4 on macrophages increases the production of IL-10 and C-C Motif Chemokine Ligand 22 (CCL22) that promote the expansion and activation of CD4+CD25^high^FOXP3+ Tregs, suppressing immune responses against the tumor. The inhibition of TLR4 signaling in macrophages reduced the number and activity of Tregs and slowed tumor growth in mice with HCC [132]. Along the same line, the TLR4/C-X-C motif chemokine ligand (CXCL)10/C-X-C motif chemokine receptor (CXCR3) signaling axis, regulates Treg infiltration after liver transplantation. Specifically, the activation of TLR4 by LPS induces the secretion of CXCL10 by Kupffer cells. CXCL10 further binds to CXCR3 on Tregs promoting their migration to the liver and facilitating the establishment of an immunosuppressive microenvironment that promotes tumor growth and recurrence after transplantation [133]. Finally, Wang et al. demonstrated that TLR4 signaling is required for the metabolic reprogramming of naïve CD4+ T-cells by NETs. Despite the lower absolute number of CD4+ T-cells in NASH, they documented that there was a selective increase in Tregs. The depletion of Tregs had a significant inhibitory effect on NASH-related HCC initiation and progression. Additionally, there was noted a positive correlation between increased hepatic Treg levels and NETs. RNA sequencing data revealed that NETs have an impact on gene expression profiles in naïve CD4+ T-cells, particularly those genes involved in mitochondrial oxidative phosphorylation. NETs promoted the differentiation into Tregs, facilitating mitochondrial respiration. TLR4 was needed for metabolic reprogramming of naïve CD4+ T-cells by NETs. In vivo blockade of NETs using *Pad4*−/− mice or DNase I treatment reduced Treg activity. Targeting this process could provide a way to prevent liver cancer in patients with NASH [134].

Xiao et al. further demonstrated that TLR4 signaling exerts significant influence in the regulation of B cells [135], as they reported that PD-1^high^ B cells were significantly enriched in tumor tissues compared to blood, and expressed high levels of the immunosuppressive IL-10, TGF-β, and PD-L1, inhibiting the activation and proliferation of T-cells in vitro. They further demonstrated in vivo that the depletion of PD-1^high^ B cells in the HCC mouse model resulted in reduced tumor growth and improved OS. The generation of the PD-1^high^ B cell population is dependent on TLR4-mediated upregulation of B-cell lymphoma 6 (BCL6), which is reversed by the IL-4-induced phosphorylation of STAT6 [135].

#### 4.2.2. Regulation of DCs by TLR4 Signaling

DCs play a critical role in anti-cancer immunity. DCs are antigen-presenting cells that capture and process antigens and then present them to T cells to initiate an immune response. In cancer, however, the TME can inhibit the function of DCs, leading to a failure of the immune system to recognize and attack the tumor. Tumor cells may release factors that suppress DC function, such as TGF-β and IL-10, or hinder their differentiation and maturation, such as VEGF, ultimately leading to the prevention of T cell activation and ineffective immune responses [136]. Yamamoto et al. demonstrated that the alpha-fetoprotein (AFP) inhibited the function of DCs, highlighting that AFP suppressed the production of pro-inflammatory cytokines IL-12p35 and IL-12p40 by DCs and impaired their ability to activate NK cells, which are important immune effector cells [137]. These findings indicated that AFP could potentially impede the TLR-3 or TLR-4 signaling pathway, leading to the inhibition of mRNA translation of the *IL-12* gene [137].

#### 4.2.3. Regulation of Neutrophils by TLR4 Signaling

In HCC, neutrophils can have both tumor-promoting and tumor-inhibiting effects. On the one hand, neutrophils can release various pro-inflammatory cytokines and chemokines that promote tumor growth, invasion, and angiogenesis, and suppress the activity of immune cells that are responsible for tumor surveillance, while, οn the other hand, they can demonstrate anti-tumor effects, as they can directly kill tumor cells by releasing toxic molecules such as ROS and NETs [67]. Neutrophils can also promote the activation of cytotoxic cells to potentiate tumor immunity [68].

Recent evidence implicates TLR4 signaling in the regulation of neutrophil function in HCC. TLR4 signaling plays a critical role in promoting the formation of NETs in HCC cells, which intriguingly contributes to the promotion of tumor growth and metastasis. The activation of TLR4/NF-κBp65/COX-2 signaling in HCC cells trapped by NETs promotes angiogenesis and tumor cell migration, inducing a pro-metastatic phenotype. As mentioned above, DNAse1 inhibitors, hydroxychloroquine (HCQ), and COX-2 inhibitors could suppress the induction of this pro-inflammatory response [64]. In the same direction, Zhan et al. proposed a mechanism whereby HBV infection induces the expression of S100A9, a calcium-binding protein, which activates the TLR4/RAGE signaling pathway in CD66b+ neutrophils leading to the production of ROS and the release of NETs. Overall, the elevated levels of NETs facilitate tumor growth and metastasis, promoting angiogenesis and inducing an immunosuppressive microenvironment [65].

#### 4.2.4. Regulation of MDSCs by TLR4 Signaling

MDSCs are a heterogeneous population of immature myeloid cells that play a critical role in the immune evasion of tumors, including HCC [138]. MDSCs exert their suppressive activity through various mechanisms, including the production of ROS, arginase, and cytokines that inhibit the function of T cells, NK cells, and DCs [139]. MDSCs also promote the expansion of Tregs, which further suppress the immune response [140] and limit the efficacy of checkpoint blockade [141].

Accumulating evidence implicates TLR4 signaling in the regulation of MDSCs in HCC. In a study of 331 patients with HCC who underwent liver transplantation, those with a graft weight ratio (GWR) less than 60%, which is the weight of the liver graft divided by the standard liver weight of the recipient, had a higher risk of tumor recurrence compared to those with a GWR of 60% or more. Additionally, patients with a GWR of less than 60% or tumor recurrence had significantly elevated levels of MDSCs and CXCL10/TLR4. In a model of hepatic I/R injury plus major hepatectomy (IRH) in knockout mice lacking *Cxcl10* or *Tlr4* genes, monocytic MDSCs were significantly reduced while granulocytic MDSCs were not affected. Interestingly, deficiency of CXCL10 led to a decrease in the accumulation of TLR4+ monocytic MDSCs, and CXCL10 increased MDSC mobilization in the presence of TLR4. Moreover, matrix metallopeptidase (MMP) 14 was found to be the crucial molecule connecting CXCL10/TLR4 signaling and MDSC mobilization. Overall, evidence suggested that during the acute phase of injury, monocytic MDSCs were mobilized and attracted to the liver graft, which subsequently led to the promotion of HCC recurrence following transplantation. To reduce the likelihood of liver tumor recurrence after transplantation, a possible therapeutic approach could be the targeting of MDSC mobilization through CXCL10/TLR4/MMP14 signaling [142]. Additionally, Hu et al. demonstrated that MDSCs accumulated in the liver during HCC development and contributed to tumor progression by suppressing the function of DCs. Specifically, the MyD88-NF-κB signaling in MDSCs caused IL-10 secretion, which led to the downregulation of IL-12 secretion by DCs, resulting in the suppression of T cell activation and proliferation due to the lack of co-stimulatory and major histocompatibility (MHC) class II molecules. In summary, TLR4 plays a crucial role in the upregulation and expansion of MDSCs, which contribute to HCC development by impairing the function of DCs and suppressing the activation of tumor-specific T cells [143].

#### 4.2.5. Regulation of Macrophages by TLR4 Signaling

Macrophages play a critical role in the development and progression of HCC. Macrophages can be classified into two main subtypes: M1 and M2 macrophages [144]. M1 macrophages are involved in the initiation of the immune response and are characterized by the production of pro-inflammatory cytokines such as IL-1β, IL-6, and TNFα, while M2 macrophages are involved in the resolution of inflammation and tissue repair and are characterized by the production of anti-inflammatory cytokines such as IL-10 and TGF-β [145]. In HCC, macrophages can promote or inhibit tumor growth depending on their phenotype and activation status. M2-like macrophages, which are characterized by the expression of markers CD163 and CD206, have been shown to promote tumor growth and metastasis by producing growth factors and cytokines that stimulate angiogenesis and immunosuppression; M1-like macrophages, on the other hand, which are characterized by the expression of markers CD86 and HLA-DR, have been shown to exhibit anti-tumor activity by producing cytokines that activate T cells and induce tumor cell apoptosis. The above are better analyzed elsewhere [144] and a more in-depth discussion goes beyond the scope of our review.

TLR4 signaling in macrophages is of paramount importance for the development of steatohepatitis-induced HCC [146]. Miura et al. demonstrated in a mouse model of steatohepatitis-related HCC that mice lacking TLR4 had significantly fewer liver tumors than the control mice. They further showed that TLR4-expressing macrophages contributed to the development of HCC in *Pten*-deficient (*Pten*^Δ*hep*^) mice by promoting the production of pro-inflammatory cytokines such as IL-6 and TNFα, as well as ROS. This is an important finding, as it suggests a potential therapeutic target for both the prevention and treatment of HCC in patients with NASH [146]. The serine/threonine kinase 4 (STK4) protein is a critical regulator of TLR pathways in macrophages. Specifically, STK4 inhibits TLR4 signaling in macrophages, leading to a reduction in the production of pro-inflammatory cytokines [147]. M1 polarization could be a promising therapeutic strategy against HCC, and mounting evidence in this direction has started to emerge. Astragaloside IV inhibits HCC progression by modulating macrophage polarization, and Min et al. showed that Astragaloside IV through the downregulation of the TLR4/NF-κB/STAT3 signaling pathway stimulates the aggregation of M1 macrophages and hinders M2 macrophages in vitro and in vivo [148]. Furthermore, Pan et al. reported that combining TLR4 agonists with glucocorticoid-induced TNF receptor (GITR) agonists can reverse the M2 polarization of macrophages in HCC and enhance anti-tumor immunity. In murine HCC models, treatment with the LPS and DTA-1 (an agonistic GITR antibody) combination therapy significantly inhibited tumor growth compared to treatment with either LPS or DTA-1 alone. The combination therapy also increased the infiltration of M1 macrophages and CD8+ T cells in the TME, indicating enhanced anti-tumor immunity [149]. Regarding the role of TAMs in HCC, Yao et al. showed that TAMs promoted the migration and EMT of HCC cells, which was mediated by the TLR4/STAT3 signaling pathway. They also found that inhibition of this pathway could reduce the effects of TAMs on HCC cells and thus may represent a potential therapeutic target for HCC [80]. Evidence from another study also suggested that HCC-derived extracellular vehicles (EVs) containing prostate androgen-regulated transcript 1 (PART1), a long non-coding RNA (lncRNA), promoted the M2 polarization of macrophages and enhanced HCC cell proliferation and migration either through the PI3K/AKT or the Janus kinase (JAK)/STAT signaling pathway, suggesting that targeting these EVs may be a potential therapeutic strategy for HCC by inhibiting macrophage polarization and disrupting the tumor-promoting microenvironment [94]. Further research is necessary to validate these findings and determine the potential clinical implications of targeting these signaling pathways.

#### 4.2.6. Regulation of Fibroblasts by TLR4 Signaling

HCC mostly develops within a fibrotic microenvironment, where HSCs and carcinoma-associated fibroblasts (CAFs) play a crucial role in HCC progression [150]. CAFs are a subset of fibroblasts that are present in the TME of HCC, and they partake in promoting tumor growth, progression, and metastasis through a variety of mechanisms. One of the primary functions of CAFs in HCC TME is the production and deposition of ECM components, creating a supportive environment for tumor growth and invasion [151]. Moreover, CAFs can also regulate angiogenesis by producing various pro-angiogenic factors, including VEGF, FGF, and PDGF, which can stimulate endothelial cell proliferation and migration, leading to the formation of new blood vessels. Recent studies have also shown that CAFs in HCC TME can promote immunosuppression and escape from immune surveillance, and that they can produce factors that inhibit the activity of immune cells, including T cells, NK cells, and DCs, leading to a more permissive environment for tumor growth [152]. Increasingly compelling evidence suggests that TLR4 signaling is implicated in the regulation of fibroblast/CAF-HCC cell interconnection.

Song et al. aimed to investigate the role of TLR4 signaling in the development of liver fibrosis and cancer in mice with hepatocyte-specific *Tak1* deletion [105]. In more detail, this deletion leads to liver inflammation and fibrosis, which can eventually progress to liver cancer. They also reported that TLR4 and TLR9 activation is increased in the liver tissue of mice with *Tak1* deletion, and that this activation is necessary for the development of fibrosis and cancer [105]. Similarly, Lie et al. proposed that LPS-induced differentiation of hepatic progenitor cells into myofibroblasts contributes to the development of HCC [153]. In more detail, they documented that LPS exposure led to the differentiation of hepatic progenitor cells into myofibroblasts in vitro. In vivo, however, they reported that mice treated with LPS developed a more severe liver injury, increased myofibroblast activation, and a higher incidence of HCC [153]. In addition, Lu et al. suggested that LPS promotes angiogenesis in HCC by stimulating HSC activation via the TLR4 pathway [117], and they showed that LPS treatment increased HSC activation, as demonstrated by the increased expression of α-SMA and collagen I in HSCs. LPS treatment also led to increased angiogenesis in HCC, as evidenced by the increased microvessel density and VEGF expression [117]. Loh et al. demonstrated that TLR4 signaling is critical in maintaining the immature phenotype of tumor-initiating cells (TIC) [85]. Follistatin-like 1 (FSTL1) has been identified as a pro-inflammatory mediator in various fibrosis-related and inflammatory diseases. FSTL1 lineage cells give rise to myofibroblasts, and high FSTL1 expression in fibroblast activation protein (FAP)+ fibroblasts, was significantly associated with advanced disease in HCC patients. Treatment of Hep3B, HepG2 cells, and patient-derived 3D organoids with recombinant FSTL1 or conditioned medium collected from HSCs or cells overexpressing FSTL1 promoted HCC growth and metastasis by binding to the TLR4 receptor and activating AKT/mammalian target of rapamycin (mTOR)/4E-binding protein 1 (4EBP1) signaling. Blocking FSTL1 in a preclinical mouse model mitigated HCC malignancy and metastasis, sensitized HCC tumors to sorafenib, prolonged survival, and eradicated the TIC subset. These findings suggested that TLR4 and FSTL1 may serve as novel diagnostic/prognostic biomarkers and therapeutic targets for HCC [85]. Ding et al. further stated that HCC cells acquire a pro-inflammatory and stem cell phenotype which is thought to play a crucial role in tumor initiation and progression [154]. Specifically, they found that the presence of MRC-5 cells induced the expression of cancer stem cell markers in HCC cancer cells, including CD133 and Oct-4, and also upregulated the expression of several adhesion molecules, including intercellular adhesion molecule 1 (ICAM-1), vascular cell adhesion molecule 1 (VCAM-1), and E-selectin [154]. Finally, Shen et al. provided evidence that radiation exposure can promote the invasive potential of HCC cells through TLR4-mediated activation of HSCs [155]. They reported that co-culture of MHCC97-L, Hep-3B, and LM3 cells with irradiated HSCs, significantly improved the invasion potential of HCC cells. Moreover, co-culture with irradiated HCC cells increased the invasive capacity of HSCs associated with activation phenotype and upregulated TLR4 signaling, linked to enhanced ICAM-1, laminin receptor (LR), IL-6, and CX3CL1 expression, and decreased the expression of toll-interacting proteins. Epigallocatechin-3-gallate (EGCG) was also found to inhibit TLR4 signal transduction by binding to LR [155]. These findings offer insight into the mechanism of post-radiotherapy recurrence and metastasis of liver cancer and may be of use in enhancing the therapeutic efficacy of liver cancer stereotactic body radiation therapy. (Figure 3).

### 4.3. Targeting TLR4 Signaling to Enhance PD-1 Blockade

Several ICI combinations have shown promising results in the treatment of HCC, including nivolumab (anti-PD-1) plus ipilimumab (anti-CTLA-4), pembrolizumab (anti-PD-1) plus lenvatinib (TKI), which did not manage to outperform lenvatinib in OS and progression-free survival (PFS), and atezolizumab plus bevacizumab, which improved PFS and OS in clinical trials and comprise the mainstay of HCC treatment [156]. Nivolumab plus cabozantinib (TKI), pembrolizumab plus sorafenib (TKI), and atezolizumab plus cabozantinib are currently under clinical trials, while a new approach to strengthening the potential of anti-tumor immunotherapies involves combining immunotherapeutic agents with epigenetic drugs, such as DNMT inhibitors (DNMTi) and HDAC inhibitors (HDACi). The utilization of these inhibitors can lower PD-L1 expression and increase the effectiveness of PD-L1-blocking antibodies against tumors [157].

The pyroptosis signaling pathway, a type of programmed cell death that triggers inflammation, comprises another potential therapeutic target to enhance the immunotherapy response [158]. Gasdermin E in tumor cells increases the infiltration of immune cells into the TME, such as T and NK cells, inducing pyroptosis. Inflammation leads to the release of tumor antigens, which can then activate the immune system’s anti-tumor response [159]. Lv et al. suggested that targeting the protein gasdermin D (GSDMD) in HCC cells can sensitize them to anti-PD-1 therapy, which was achieved by activating the cyclic guanosine monophosphate–adenosine monophosphate synthase (cGAS) pathway and downregulating PD-L1 expression [160]. They also provided evidence that targeting GSDMD in HCC cells leads to the release of DNA from dying cells, activating the cGAS pathway [160]. The cGAS pathway is activated by the binding of cytoplasmic DNA to the cGAS enzyme, which produces the second messenger cGAMP (cyclic GMP-AMP). cGAMP then binds to the protein stimulator of interferon genes (STING) which triggers the production of type I IFN. The HMGB1/TLR4/caspase- 1 signaling activated the cGAS pathway, leading to the release of DNA from cells and thus creating a positive feedback of continuous activation [160]. Thus, an effective treatment option for HCC patients with upregulated GSDMD could be a combination therapy with an anti-PD-1 agent, a GSDMD inhibitor, alongside endogenous or exogenous ligands of TLR4 [160].

Besides the activation of the TLR4 signaling pathway in DCs driving the generation of an HCC vaccine [125], Lmdd-MPFG has demonstrated additional immune-modifying effects and the potential to exhibit further effective therapeutic outcomes. Xu et al. explored the combination of Lmdd-MPFG with an anti-PD-1, demonstrating that Lmdd-MPFG enhanced the expression of PD-L1 in HCC cells and sensitized local T cells to respond to anti-PD-1. The mechanism involves the activation of the TLR2/MyD88/NF-κB pathway in TAMs by the Lmdd-MPFG vaccine, recruiting p62 to activate the pathway of autophagy, leading to a shift of TAMs from the M2 polarization to an M1 state, altering the cytokine profile to an antitumor one in the TME [161]. This change restored T cell reactivity to the anti-PD-1 blockade, providing a promising new strategy for HCC treatment.

## 5. Discussion

HCC accounts for almost 90% of all cases, while systemic therapy is administered to around half of HCC patients, especially in the advanced stages of the disease. Preliminary data had implicated NF-kB signaling in HCC carcinogenesis. Pikarsky et al. demonstrated that suppressing NF-kB inhibition through anti-TNFa treatment or induction of IkB super-repressor in later stages of tumor development resulted in apoptosis of transformed hepatocytes and prevented progression to HCC. They suggested that NF-kappaB is essential for promoting inflammation-associated cancer and could be a potential target for cancer prevention in chronic inflammatory diseases [162]. Maeda et al. found that mice lacking IKKb only in hepatocytes (Ikkbeta (Deltahep) mice) had a significant increase in DEN-driven HCC. This was due to an increase in the production of ROS, activation of the JNK pathway, and death of hepatocytes, which led to the compensatory proliferation of surviving hepatocytes. The administration of antioxidants prevented excessive carcinogenesis by suppressing prolonged JNK activation and compensatory proliferation. Additionally, mice lacking IKKb in both hepatocytes and hematopoietic-derived Kupffer cells had reduced hepatocarcinogenesis due to decreased hepatocyte regeneration and diminished induction of hepatomitogens, which was not observed in Ikkbeta (Deltahep) mice. The authors suggested that IKKb orchestrates inflammatory crosstalk between hepatocytes and hematopoietic-derived cells that promotes chemical hepatocarcinogenesis [163].

Immuno-oncology has revolutionized cancer treatment and demonstrated robust and long-lasting anti-tumor activity across various malignancies, including HCC. Currently, the first-line therapy for advanced HCC involves immune checkpoint inhibition with atezolizumab and bevacizumab [120], and real-world data about its efficacy have started to emerge. Himmelsbach et al. evaluated the demographics, OS, and adverse events of 66 patients with advanced HCC who received atezo/beva as first-line treatment at four cancer centers in Germany and Austria between December 2018 and August 2021. Most patients had compensated cirrhosis, with Child–Pugh class B cirrhosis observed in 35% of patients and class C cirrhosis in 8% of patients. The best responses to treatment included a complete response (CR) in 11% of patients, a partial response (PR) in 18% of patients, stable disease (SD) in 33% of patients, and progressive disease in 17% of patients. The 6-month OS was 69%, the 12-month OS was 60%, and the 18-month OS was 58%. Furthermore, patients with viral hepatitis had a better prognosis than patients with HCC of non-viral etiology. Real-world PFS and OS were comparable to those reported in the pivotal IMBRAVE trial, despite the inclusion of patients with worse liver function in this study [164]. Along the same line, Fulgenzi et al., in a meta-analysis of phase III studies, concluded that the combinations of ICIs with anti-VEGF agents and double ICIs led to the greatest OS benefit, as compared to sorafenib alone, while ICI plus kinase inhibitor regimens were associated with greater PFS benefit at the cost of higher toxicity rates [156]. Gut microbiota has become an increasingly significant factor that affects anticancer immunity and the response to immunotherapy [165]. Eng et al. conducted a population-based study to investigate the effects of antibiotic treatment on the efficacy of immunotherapy in a large group of 2737 patients who received immune checkpoint blockade. They reported that exposure to antibiotics up to one year before checkpoint blockade may negatively impact outcomes, with fluoroquinolones showing the strongest association with reduced survival. These findings have important implications for clinical practice and suggest that therapies aimed at modulating the microbiome may be a promising approach to enhancing immunotherapy responses in patients [166].

This aforementioned progress is a result of substantial endeavors to comprehend networks and interactions within the TME [167,168], and extensive translational research conducted with the purpose of discovering biomarkers. Despite the fact that ICIs have revolutionized the management of HCC, only a small percentage of patients, approximately 20%, respond adequately to ICIs, highlighting the need to identify biomarkers that can accurately predict individual patients who may benefit from this treatment [169]. Sia et al., upon analyzing HCC samples from 956 patients, discovered that nearly 25% of them displayed markers indicative of an inflammatory response. They further identified two subclasses, one exhibiting adaptive immune responses, and another displaying exhausted immune responses, suggesting that certain HCCs may be more susceptible to therapeutic modalities that target regulatory pathways in T cells, such as PD-L1, PD-1, and TGF-β1 inhibitors [170], constituting the foundation for the implementation of ICIs in HCC. Montironi et al. further provided a thorough characterization of the HCC immunological classes [169], as they identified a cluster of tumors harboring immune traits, referred to as the immune-like subclass, which resembled the previously described immune subclass but was not captured by the immune signature. *CTNNB1* mutations are associated with two distinct types of immune activation. Most tumors with activation of the Wnt-β catenin pathway show features of immune exclusion and immune cell paucity, but a minority of about 15–30% of tumors with activation of this pathway express features of inflammation and immune activation and belong to the inflamed subclass. Importantly, these tumors overexpress CCL5 and CCL4 and have enhanced CD8+T cell infiltration. They have also developed a novel inflamed signature that can identify these tumors with an accuracy of approximately 90%, and they have further validated it in three additional cohorts [169]. In the same direction, Haber et al. generated an 11-gene signature that holds promise in predicting response to anti-PD1 treatment and survival [171]. Out of the 83 patients with transcriptomic data, 28 received frontline ICI, while 55 were treated with TKI in the second or third line, and patients who responded to frontline treatment exhibited upregulation of interferon-g signaling (*STAT1*, *STAT2*, and *IRF1*) and MHCII-related genes (*HLA-DRA*, *HLA-DQA1*, and *HLA-DMA*) [171]. Finally, Zhu et al. attempted to dissect the immune TME in order to discover biomarkers of response in atezolizumab plus bevacizumab treatment [172]. They conducted molecular analyses of tumor samples from 358 patients with HCC who participated in the GO30140 phase Ib or IMbrave150 phase III trial and were treated with atezolizumab and bevacizumab, atezolizumab alone, or sorafenib. They found that patients who expressed immune markers such as PD-L1/CD274 exhibited T-effector (Teff) phenotype with intratumoral infiltration of CD8+ T cells, and demonstrated better clinical outcomes with the combination therapy. On the other hand, patients with a high Treg-to-Teff ratio and expression of oncofetal genes (*GPC3*, *AFP*) derived less clinical benefit. Finally, they reported that improved outcomes linked to combination therapy were associated with high expression of VEGFR2, Tregs, and a myeloid inflammation phenotype [172].

Given the increasing incidence of cholangiocarcinoma (CCA) worldwide, the role of TLR4 signaling in CCA requires a brief mention. CCA, an aggressive malignancy with a 5-year OS of less than 10%, is the second most common primary hepatobiliary carcinoma, with an increasing global incidence over the last 30 years, from 0.1 to 0.6 cases per 100,000 population, while novel molecular alterations have been recently identified in patients with CCA, and the potential for targeted therapy is currently under the microscope [173]. The key role of the SNHG1/miR-140/TLR4/NF-κB signaling axis in CCA tumorigenesis and progression has been established, indicating that TLR4 expression promotes cell proliferation and angiogenesis and inhibits apoptosis, stimulating tumor invasion and metastasis [174]. Similarly, another study demonstrated that S100 calcium-binding protein A8 (S100A8) had an important role in facilitating CCA cell migration and metastasis via upregulation of VEGF expression by activating the TLR4/NF-κB pathway, providing a potential novel target for CCA treatment [175]. Finally, Xiao et al. provided evidence that the circular RNA Circ_0000591, served as endogenous RNA for miR-326 to promote the progression of CCA via the TLR4/MyD88/NF-κB/IL6 axis, enhancing our knowledge of potential molecular mechanisms involved in the malignant progression of CCA [176]. Therapeutic possibilities for patients with CCA may increase in number as our knowledge regarding the tumor microenvironment and its impact on tumor growth increases.

Despite the numerous advancements in the management of HCC, we highlight the need for further research and the development of personalized therapies, in order to optimize the treatment outcomes of patients with HCC, while immunotherapeutic strategies such as oncolytic viro-immunotherapy and adoptive T cell transfer are under investigation. Key challenges in the field include predefining the role of immunotherapy in the earlier stages of HCC, the exploration of combinational strategies targeting the TME with ICIs, and, last but not least, developing other treatment options for patients who do not respond to current immunotherapies [177].

## 6. Conclusions

Cancer research has progressed from being tumor-cell-focused towards having a broader system-wide perspective, encompassing dynamic host responses, tumor microenvironment, and inter-cellular signaling. Tumor cells are part of the complex tissue-specific ecosystem termed the tumor microenvironment, which is comprised of various cell types, signaling molecules, the extracellular matrix, and the microbiome, while the acknowledgment that the TME can both hinder and promote tumorigenesis has necessitated further research on TME biology. To shed some light on the complex nature of the TME, a better understanding of the communication between cancerous and surrounding non-cancerous cells is crucial.

In this review, we delved into the rationale, mechanistic basis, and preclinical evidence supporting the role of TLR4 signaling in HCC immunotherapy. However, it is important to note that the role of TLR4 signaling in cancer is complex and context-dependent, and its nonselective inhibition may exert both positive and negative effects on the immune response. Further research is required to fully understand the role of TLR4 signaling on HCC immunotherapy in order to accurately manipulate the receptor into either enhancing or inhibiting its signal within the TME, which would lead to the development of optimal and individualized treatment strategies for patients with HCC.

## Figures and Tables

**Figure 1 cancers-15-02795-f001:**
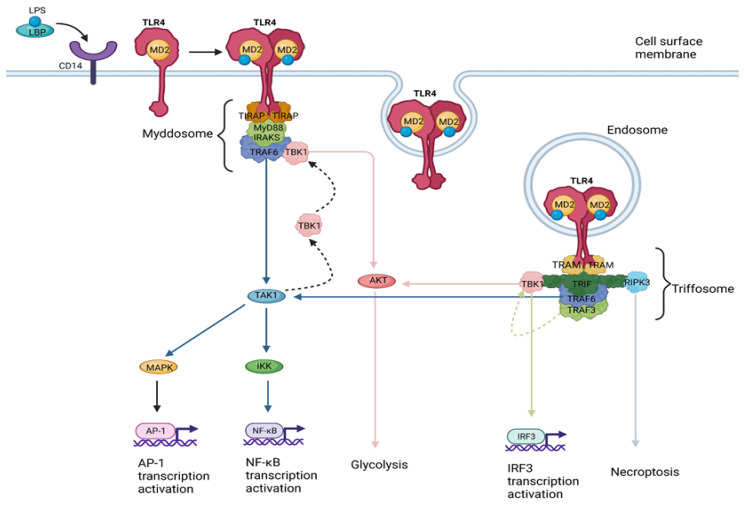
Summary of the fundamental parts of TLR4 signaling. The TLR4 signaling pathway involves several key molecules including adaptor proteins (TIRAP and TRAM), kinases (PI3K/Akt, MAPK, and IKK), and transcription factors (AP-1, NF-kB, and IRF3). These molecules work together to activate the downstream signaling transduction and regulate the expression of target genes. Created with Biorender.com.

**Figure 2 cancers-15-02795-f002:**
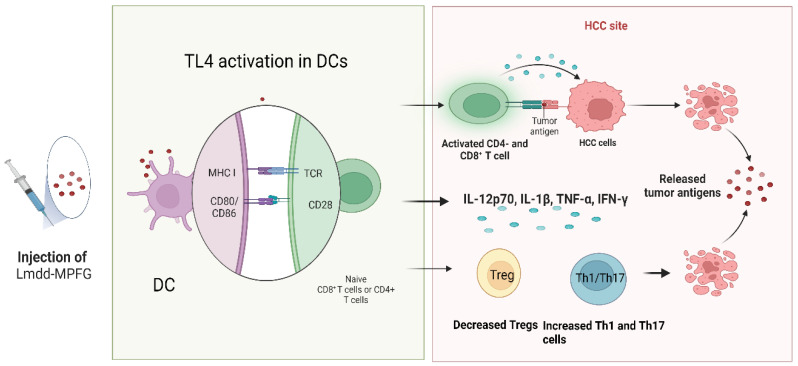
Basic principles of Lmdd-MPFG function. The vaccine uses an attenuated strain of Listeria modified to express GPC3 found on the surface of liver cancer cells. The vaccine activates DCs, leading to increased production of cytokines (IL-12p70, IL-1β, TNF-α, and IFN-γ), enhanced ability to prime T cells, and a decrease in Treg cells with an increase in Th1 and Th17 cells, promoting antitumor immunity. Created with Biorender.com.

**Figure 3 cancers-15-02795-f003:**
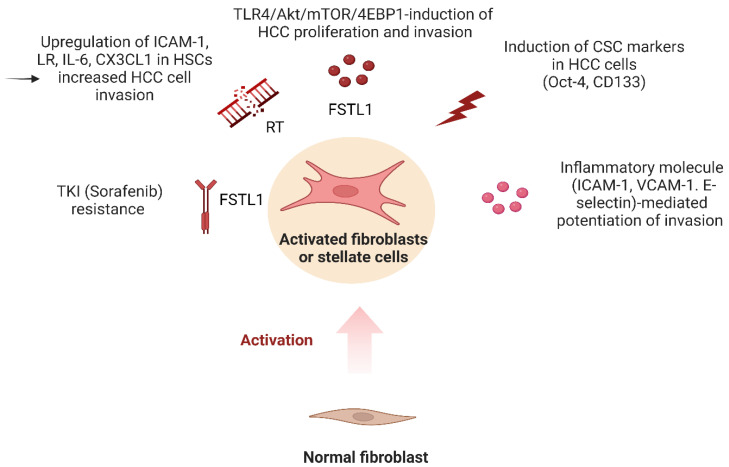
The role of carcinoma-associated fibroblasts (CAFs) and TLR4 signaling in the development and progression of HCC. TLR4 activation is necessary for the development of fibrosis and cancer. CAFs promote tumor growth, angiogenesis, and immunosuppression by producing extracellular matrix components and pro-angiogenic factors. TLR4 signaling is critical in maintaining the immature phenotype of tumor-initiating cells and is implicated in the regulation of fibroblast/CAF-HCC cell interconnection. Created with Biorender.com.

**Table 1 cancers-15-02795-t001:** Summary of the mechanisms of chemoresistance in HCC.

Mechanism of Chemoresistance (MOC)	Cellular Function
MOC-1a	Uptake carriers
MOC-1b	Export pumps
MOC-2	Drug metabolism
MOC-3	Therapeutic/Drug targets
MOC-4	DNA repair
MOC-5a	Apoptosis potentiation
MOC-5b	Pro-survival signaling
MOC-6	Tumor microenvironment modification
MOC-7	EMT regulation

**Table 2 cancers-15-02795-t002:** Brief summary of the influence of the TLR4 signaling pathway in drug resistance in HCC.

Molecule	Chemotherapeutic	Cell Lines	Signaling Pathway	Outcomes	References
miR-145	5-fluorouracil (5-FU)	SNU449 cells, SNU449/5-FU and Huh7 cells, Huh7/5-FU cells	TLR4/MyD88	Drug sensitivity	[84]
				Upregulation of caspase3, caspase9, Bax	
				Downregulation of Bcl-2	
Follistatin-like 1 (FSTL1)-neutralizing antibodies	Sorafenib	MHCC97L cells	TLR4/AKT /mTOR/4EBP1	Drug sensitivity	[85]
COX6A2 and FAO inhibition	Sorafenib	CD133+/CD49f+ TICs from human HCCs	TLR4/E2F1/NANOG	Drug sensitivity	[81]
				Increased cytochrome release	
LPS	Doxorubicin	SMMC-7721, Hep-3B cells	TLR4/AKT/SOX2	Drug resistance	[86]
				Enhancement of apoptosis	

**Table 3 cancers-15-02795-t003:** Summary of the effects of epigenetic targeting of TLR4 signaling in HCC.

Molecule	Cell Lines	Signaling Pathways	Outcome	References
miR-145	SNU449 cells, SNU449/5-FU and Huh7 cells, Huh7 /5-FU cells	TLR4/MyD88	Increase 5-FU sensitivity	[84]
			Upregulation of Caspase3, Caspase9, Bax	
			Downregulation of Bcl-2	
histone	Hepa1-6 cells, HuH7 cells and TLR4−/− mice or TLR4 knockdown cells”	TLR4/ERK/NF-κB/CCL9-10	Enhancement of migration/invasion and metastasis	[91]
miR-122 mimic	Hep3B and MHCC97H cells	PI3K/Akt/NF-κB	Immune escape	[92]
			Downregulation of VEGF, IL-6, COX-2, PGE2, MMP-9	
miR-122 mimic	HepG2 and Huh7 cells	Linkage to 3′ UTR of TLR4	Downregulation of proliferation	[93]
			Downregulation of TNFα and IL-6	
lncRNA PART1 or miR-372-3p blocker	HB611, Huh7, HCCLM3, Bel 7405, THLE-2 cells and human monocytes THP 1 cells	miR-372-3p/TLR4 axis	M2 TAM polarization	[94]
			Enhancement of proliferation, migration, EMT	
let-7g miRNA	PLC5 cells	TLR4/LIN28A/let-7g	Positive feedback loop enhancing stemness and proliferation	[95]
			Downregulation of *TLR4* mRNA	

**Table 4 cancers-15-02795-t004:** Brief presentation of the effects of TLR4 activation or blockage in HCC proliferation and/or apoptosis.

Molecule	Cell Lines/Animals	Signaling Pathway	Outcome	Refs
LPS	antibiotics + DEN treated rats, TLR4−/− mice, TLR4-mutant mice	LPS/TLR4/NF-κB in HCC cells and Kuppfer cells	TLR4/NF-κB signaling in Kuppfer cells: IL-6, TNFα	[98]
			TLR4/NF-κB signaling in Kuppfer cells: upregulation of antioxidant enzyme superoxide dismutase (SOD) and anti-apoptotic molecules (A20, Bcl-xl)	
Desacetyluvaricin (DES)	HepG2.2.15 cells as controls, DES, cisplatin	LPS/P53	Increased TLR4 expression in DES vs. control (71.94% vs. 37.16%, *p* < 0.05)	[99]
			Increased P53 expression (32.6% vs. 3.3%, *p* < 0.05)	
Baishouwu extracts	Baishouwu-treated /non-treated rats	TLR4/MyD88/NF-κB in myofibroblasts and HCCs	Downregulation of DEN-related liver inflammation, fibrosis, HCC	[100]
CXC195	HepG2 cells	TLR4/NF-κB TLR4/MAPK TLR4/MyD88/TAK1	Downregulation of TLR4 expression	[101]
			Partial downregulation of cell cycle (increase cells in G0/G1 phase)	
			Downregulation of inflammatory cascade: IL-6, CCL-2, CCL-22, EGFR	
Adamantane byproducts	Thioacetamide (TAA)-induced HCC, Hep-G2 cells, HCC-DOXO rats	TLR4/MyD88/NF-κB	Downregulation of TLR4, MyD88, TRAF-6, p-NF-κB, p65	[102]
			Downregulation of liver fibrosis	
ginsenoside Rk3	dimethyl nitrosamine-, CCl4-mediated HCC mouse model	LPS/TLR4 signaling	Inhibition of liver damage, fibrosis, and tumor burden	[103]
			Induction of apoptosis: increased Bax, decreased Bcl-2 expression	
taxifolin/alogliptin	diethyl nitrosamine-related HCC in rats		Increased survival rate	[104]
			Marked upregulation of NRF2 expression	
			Downregulation of IL-1a, TLR4 expression	
			Upregulation of beclin-1, caspase-3, caspase-9, JNK	
USP13	SK-HEP-1, HepG2, Huh7, Hep3B	TLR4/MyD88/NF-κB	USP13 silencing: downregulation of TLR4 signaling	[59]
BMS345541 (IKK inhibitor), WP1066 (STAT3 inhibitor)	Huh7, PLC5 cells	positive feedback loop: TLR4/LIN28A/let-7g TLR4/STAT3/NF-κB TLR4/PI3K/Akt	Upregulation of TLR4, IL-6 and CCL2 expression	[95]
			Enhanced survival and proliferation	
	double knockout mice lacking Tak1 in liver	TLR4/MyD88/TNFR	Liver damage, macrophage infiltration, collagen degradation and tumor growth	[105]
L. Plantarum	Wistar rats and/or L. Plantarum-, TAA-treated	TLR4/CXCL9/PREX-2	Increased mean hep par-1 density	[106]
Sorafenib plus Fluvastatin	HepG2, SK-Hep-1 cells, LX-2 cells, DEN-treated rats	HCC cells: TLR4/MAPK Hepatic stellate cells: TLR4/NF-κB/SDF1α/MAPK	HCC cells: Decreased proliferation, Increased apoptosis	[107]
			Hepatic stellate cells: Decreased activation, fibrosis	
T2BP	SK-Hep1, HepG2 cells/male NOD/SCID mice	TLR4/NF-κB	p53-independent activation of caspase 3, 8	[108]
			Induction of p53-mediated cell cycle arrest	
miR-122 mimic	Hep3B and MHCC97H cells	PI3K/Akt/NF-κB	Immune escape	[92]
			Downregulation of VEGF, IL-6, COX-2, PGE2, and MMP-9	
PRL	male and female C3H/HeN mice, Hep3B, HuH7, PH5CH8, HepG2, and HuH6 cells	PRL/PRLR signaling	PRL/PRLR-mediated downregulation of IL-1β, TNFα and TLR4 signaling (ubiquitination of “trafosome”) leading to inactivation of c-myc	[109]
Celastrol plus metformin	diethylnitrosamine (DEN)-treated BALB/c mice	Drug-mediated downregulation of TNFR, TLR4 mRNA expression leading to NF-κB inactivation	Enhancement of apoptosis	[110]
			Downregulation of pyroptosis	
LPS (TLR4 activator)	HL-7702, PLC/PRF/5, HepG2	LPS/TLR4/MAPK LPS/TLR4/ERK/JNK LPS/TLR4/p38	Positive correlation of TLR4 and Ki-67	[111]
			Increased Bax translocation to mitochondria	
			p38 downregulation induces HCC proliferation	
Chitosan-coated iron oxide nanocomposite (Fe3O4/Cs)	Fe3O4/Cs- and/or DEN-treated male albino rats	PI3K/Akt/mTOR, MAPK (ERK, JNK, P38)	Fe3O4/Cs-mediated downregulation of TLR4 expression	[112]
			Downregulation of both PI3K/Akt/mTOR, MAPK signaling	
			Upregulation of caspase-3	
Extracellular HSP70-peptide complexes	HepG2 cells	TLR2-TLR4/JNK1/2/MAPK	Upregulation of TLR2- and TLR4-overexpressing cells mediated by cyclin D1	[113]
TLR4	Male nude BALB/C mice/HepG2.2.15, MHCC97-H, Hep3B cells	TLR4/b-catenin	Reduction of b-catenin-mediated HCC apoptosis	[70]
			Enhancement of HCC proliferation	
TLR4	HepG2, H7402 cells /TLR4−/− mice	TLR4/COX-2/PGE2/STAT3 signaling	Upregulation of cell cycle (more cells in S phase, increased cyclin D1)	[114]
anti-IL-17A	methionine-choline deficient (MCD) or high-fat MCD (HFMCD) FGF21 knockout (KO) mice/Hepa1-6-FGF21KD, FL83B-FGF21KD, 3T3-L1 cells	free fatty acids/TLR4/NF-κB /IL-17a	Decreased tumor growth and nodule burden in FGF21 knockout (KO) mice	[115]
Dihydrotestosterone (DHT)	DEN- or CCl4-induced HCC/HepG2, H7402, Hepa1-6, and HepG2.2.15”	Androgen receptor (AR)-TLR4 cross-signaling	Enhanced Ki-67 in male wild-type mice compared to female	[72]
			Ki-67 equalization in TLR4−/− mice	

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
