# Peer review of "The Role of TLR4 in the Immunotherapy of Hepatocellular Carcinoma: Can We Teach an Old Dog New Tricks?"

_cancers, 2023, doi:10.3390/cancers15102795_

Round 1

Reviewer 1 Report

Very interesting and comprehensive review of the current literature on the involvement of TLR4 signaling in liver diseases and HCC, as well as its potential utilization as an immunotherapy target for HCC.

Specific comments:

-Fig. 2 and 3 are very complex and must be explained well in the legend (all steps).

- Cholangiocarcinoma is the second most common malignant liver tumor after HCC. In this review the role of TLR4 signaling in CCA is not mentioned, I suggest to include this topic.

Author Response

Response: Dear Reviewer,

Thank you for taking the time to review our article and for providing us with your insightful feedback. We appreciate your positive comments regarding the comprehensive review of the current literature on TLR4 signaling in liver diseases and its potential use as an immunotherapy target for HCC.

Point 1. Fig. 2 and 3 are very complex and must be explained well in the legend (all steps).

Response: We thank the reviewer for the comment. We agree that Figures 2 and 3 are quite complex, and we have explained accordingly all steps in their figure legends to improve their clarity.

Point 2. Cholangiocarcinoma is the second most common malignant liver tumor after HCC. In this review the role of TLR4 signaling in CCA is not mentioned, I suggest to include this topic.

Response: We thank the reviewer for the suggestion. We acknowledge that CCA is the second most common malignant liver tumor after HCC. However, we would like to clarify that the choice to focus solely on HCC was not based on the frequency of occurrence but rather on the specific research question we aimed to address.  HCC and CCA differ in fundamental pathophysiological aspects. The vast majority of HCCs arise in the setting of liver cirrhosis. The resulting portal hypertension leads to endotoxemia, which serves as a primary stimulus for TLR4 receptors. Our main objective was to investigate the association between TLR4 signaling and HCC and to explore its potential as a therapeutic target in HCC. Nevertheless, we appreciate your suggestion and acknowledge the importance of studying the role of TLR4 signaling in CCA. Therefore, we have added according to your suggestions, a small paragraph regarding TLR4 signaling in CCA, in section 5.

We will certainly consider a thorough review of the aforementioned topic in a future publication.

Once again, we thank you for your valuable feedback.

Reviewer 2 Report

This article "The Role of TLR4 in the Immunotherapy of Hepatocellular Carcinoma: Can We teach an Old Dog New tricks?" by Papadakos et al. is a comprehensive snapshot of TLR4 discovery, its importance in immune signaling, and the potential of TLR4-based therapeutics for Hepatocellular Carcinoma. This study provides an in-depth review of major discoveries that led to the current understanding of TLR4-mediated signaling cascades and their implications in immune cell activation. Following are the comments that favor accepting this article for publication in Cancers.

1. The authors have reviewed and referred to evidence supported by preclinical studies and explained the rationale as well as the TLR4 signaling mechanisms in detail. 

2. The authors have provided a detailed perspective and have opened up the challenges of addressing TLR4 in cancer due to the complexity of the immune response related to the TLR4 signaling pathway. 

3. The authors have opened questions that need further investigation on developing TLR4 therapies for Hepatocellular Cancer (HCC) 

4. The authors have provided a new perspective on the path of developing optimal and personalized treatment strategies for HCC patients.

Author Response

Response: Dear Reviewer,

We would like to express our sincere gratitude for your kind words regarding our article. It is truly heartening to receive such positive feedback.

Your comments on the importance of TLR4-mediated signaling cascades and the challenges associated with developing TLR4 therapies for HCC were also highly valued.

We are honored that you found our article to be a valuable contribution to the field and we hope that it will inspire further investigation and lead to the development of personalized treatment strategies for HCC patients.

Thank you once again for taking the time to review our article and for your constructive feedback.

Reviewer 3 Report

Overall, Papadakos and co-authors provide a reasonable overview of recent developments in TLR4 signaling as it relates to hepatocellular cancer. What I find missing in this manuscript is critical evaluation of the papers they cite. For example, Table 4 cites Reference 100 (Ding et al. 2019. Baiwushu extract suppresses the development of hepatocellular cancer via TLR4.MyD88/NF-kB pathway, Front. Pharm. 10: 1-14) as supporting Baiwushu extract action through the TLR4 pathway, however, the data shown in this paper are only correlational. The authors did not examine the specificity nor exclude other targets from mediating the effects of Baiwushu extract; furthermore, the authors failed to demonstrate that TLR4 activation was either necessary or sufficient for the extract's effects on DEN-induced inflammation or HCC. I would like to see this level critical analysis description of the strengths and limitations of at least some papers reviewed in this manuscript (perhaps 2 or 3 critical findings in each section).

Occasional errors in choice of preposition (for instance, using 'on' when 'in' is correct - line 109; 'on' when they mean 'to' - line 121; 'to' when it should be 'in' - line 262; 'with' when it should be 'in' - line 290) poor word choice ('applied' used inappropriately in line 162 (a better choice would be 'subjected' or 'treated with'; alternatively re-structure the sentence to read something like '...when warm ischemia was applied to mice...'); line 296 'Conclusively' should be 'In conclusion'. So there is some benefit to proofreading prior to publication. However, overall the manuscript is understandable.

Author Response

Response: Dear Reviewer,

Thank you for taking the time to review our article and for providing us with your insightful feedback. We appreciate your positive comments regarding the overview of the current literature on TLR4 signaling in liver diseases.

Point 1. What I find missing in this manuscript is critical evaluation of the papers they cite. For example, Table 4 cites Reference 100 (Ding et al. 2019. Baiwushu extract suppresses the development of hepatocellular cancer via TLR4.MyD88/NF-kB pathway, Front. Pharm. 10: 1-14) as supporting Baiwushu extract action through the TLR4 pathway, however, the data shown in this paper are only correlational. The authors did not examine the specificity nor exclude other targets from mediating the effects of Baiwushu extract; furthermore, the authors failed to demonstrate that TLR4 activation was either necessary or sufficient for the extract's effects on DEN-induced inflammation or HCC. I would like to see this level critical analysis description of the strengths and limitations of at least some papers reviewed in this manuscript (perhaps 2 or 3 critical findings in each section).

Response: Thank you for your valuable comments and suggestions. We have carefully considered your feedback and made the necessary revisions on the manuscript. Regarding your comment on the critical evaluation of the papers cited in the manuscript, we agree that it is an important aspect that can enhance the quality of the study. However, due to space constraints, it is not possible to provide an extensive presentation of each study in the table. Instead, we have summarized the key findings that support the respective sections.

Point 2. Comments on the Quality of English Language

Occasional errors in choice of preposition (for instance, using 'on' when 'in' is correct - line 109; 'on' when they mean 'to' - line 121; 'to' when it should be 'in' - line 262; 'with' when it should be 'in' - line 290) poor word choice ('applied' used inappropriately in line 162 (a better choice would be 'subjected' or 'treated with'; alternatively re-structure the sentence to read something like '...when warm ischemia was applied to mice...'); line 296 'Conclusively' should be 'In conclusion'. So, there is some benefit to proofreading prior to publication. However, overall, the manuscript is understandable.

Response: We appreciate your keen attention to detail and the constructive criticism you have provided. We have carefully proofread the manuscript and made the necessary corrections to address the grammatical errors you pointed out. Thank you once again for your time and effort in reviewing our manuscript.

Reviewer 4 Report

The authors present the study entitled "The Role of TLR4 in the Immunotherapy of Hepatocellular Carcinoma: Can We Teach an Old Dog New tricks".  The authors conclude that the role of TLR4 signaling in cancer is complex and context-dependent, and its nonselective inhibition can exert both positive and negative effects on the immune response. The authors present an extensive and very comprehensive review on the role of TLR4 in hepatocellular carcinoma immunotherapy. This review is very well structured and is of great interest. It can be appreciated how the authors have mastered the topic and reflect the most relevant points.

Author Response

(The authors gave the same response as above.)

Reviewer 5 Report

This paper addresses an important and interesting target in HCC development and therapy. Overall, the article is well organized and its presentation is good. However, some minor issues still need to be improved: (1) The author should summarize the role of TLR4 in hepatocarcinoma in related to non-resolved inflammation . (2) The author should summarize the role of TLR4 in hepatocelluar cell de-differentiation or differentiation. (3)In the section 3.1, the author summarized the “TLR4 signaling in HCC metastasis”, while in the section 3.2, the author summarized the “TLR4 signaling in HCC EMT”. Since EMT is a fundamental cellular process that modulates the metastatic capacity, it will be more concise to merge these two section together. 

There are a few typos and grammer errors in this paper. Some section numbers are not correct.

Author Response

Response: Dear Reviewer,

Thank you for your valuable and constructive feedback on our manuscript. We have carefully considered your suggestions and have made the necessary revisions to address the issues raised.

Point 1. The author should summarize the role of TLR4 in hepatocarcinoma in related to non-resolved inflammation.

Response: Regarding your first comment, we have included a summary of the role of TLR4 in hepatocarcinoma in relation to non-resolved inflammation in the introduction section.

Point 2. The author should summarize the role of TLR4 in hepatocellular cell de-differentiation or differentiation.

Response: For your second comment, we have provided a summary of the role of TLR4 in hepatocellular cell de-differentiation or differentiation in the discussion section.

Point 3. In the section 3.1, the author summarized the “TLR4 signaling in HCC metastasis”, while in the section 3.2, the author summarized the “TLR4 signaling in HCC EMT”. Since EMT is a fundamental cellular process that modulates the metastatic capacity, it will be more concise to merge these two sections together.

Response: We appreciate your suggestion to merge the sections 3.1 and 3.2 together as both discuss the role of TLR4 in HCC metastasis. Therefore, we have merged these two sections to provide a more concise and coherent discussion.

Point 4. Comments on the Quality of English Language

There are a few typos and grammar errors in this paper. Some section numbers are not correct.

Response: Lastly, we have carefully proofread the manuscript and corrected all typos and grammar errors. We have also ensured that all section numbers are correct.

Once again, thank you for your helpful feedback. We believe that our manuscript is now improved and ready for publication.

Round 2

Reviewer 1 Report

I have no further comments to make.